# STERLING: Self-Supervised Terrain Representation Learning from Unconstrained Robot Experience

**Haresh Karnan**[1]
haresh.miriyala@utexas.edu

**Elvin Yang**[1]
elvin.yang@utexas.edu

**Daniel Farkash**[2]
dmf248@cornell.edu

**Garett Warnell**[1,3]
garrett.a.warnell.civ@army.mil

**Joydeep Biswas**[1]
joydeepb@cs.utexas.edu

**Peter Stone**[1,4]
pstone@cs.utexas.edu

**Abstract:** *Terrain awareness*, i.e., the ability to *identify* and *distinguish* different types of terrain, is a critical ability that robots must have to succeed at autonomous off-road navigation. Current approaches that provide robots with this awareness either rely on labeled data which is expensive to collect, engineered features and cost functions that may not generalize, or expert human demonstrations which may not be available. Towards endowing robots with terrain awareness without these limitations, we introduce *Self-supervised TErrain Representation LearnING* (STERLING), a novel approach for learning terrain representations that relies solely on easy-to-collect, unconstrained (e.g., non-expert), and unlabeled robot experience, with no additional constraints on data collection. STERLING employs a novel multi-modal self-supervision objective through non-contrastive representation learning to learn relevant terrain representations for terrain-aware navigation. Through physical robot experiments in off-road environments, we evaluate STERLING features on the task of preference-aligned visual navigation and find that STERLING features perform on par with fully-supervised approaches and outperform other state-of-the-art methods with respect to preference alignment. Additionally, we perform a large-scale experiment of semi-autonomously hiking a 3-mile long trail which STERLING completes successfully with only two manual interventions, demonstrating robustness to real-world off-road conditions. Robot experiment videos and more details can be found in the appendix and the project website https://hareshkarnan.github.io/sterling/

**Keywords:** Vision-Based Navigation, Representation Learning.

## 1 Introduction

Off-road navigation is emerging as a crucial capability for autonomous mobile robots envisioned for use in a growing number of outdoor applications such as agricultural operations [1], package delivery [2], and search and rescue [3]. Endowing robots with this capability has, however, proved to be challenging and remains an active area of research.

One particularly difficult challenge in off-road autonomous navigation is that of providing the robot with *terrain awareness*, i.e., the ability to identify distinct terrain features that are relevant to a wide variety of downstream tasks (e.g., changing preferences over terrain types) [4, 5, 6, 7, 8, 9, 10, 11]. While a litany of prior work has attempted to address this challenge [12, 13, 14, 15], existing approaches typically rely on difficult-to-collect curated datasets [16, 17, 18, 19, 20] or has been

---

[1]The University of Texas at Austin, Austin, TX, USA
[2]Cornell University, Ithaca, NY, USA
[3]Army Research Laboratory, Austin, TX, USA
[4]Sony AI, North America

7th Conference on Robot Learning (CoRL 2023), Atlanta, USA.

focused on particular tasks [21, 22, 23, 24, 25, 9] and is not amenable to downstream task changes [26, 25, 7]. These limitations prevent existing approaches from appropriately scaling to the vast distribution of terrains and navigation tasks in the real world.

Toward overcoming the scalability challenges in terrain awareness, we introduce *Self-supervised TErrain Representation LearnING* (STERLING)[1], a novel approach to learning terrain representations for off-road navigation. STERLING learns an encoding function that maps high-dimensional, multi-modal sensor data to low-dimensional, terrain-aware representations that amplify differences important for navigation and attenuate differences due to extraneous factors such as changes in viewpoint and lighting. Importantly, STERLING works with easy-to-collect unconstrained and unlabeled robot data, thereby providing a scalable pathway to data collection and system improvement for the wide variety of terrain and downstream tasks that off-road robots must face.

To evaluate STERLING, we apply it to the problem of preference-aligned off-road navigation and provide a detailed comparison to existing approaches to this problem, including RCA [7], GANav [19], SE-R [8], and a fully-supervised oracle. We find that STERLING enables performance on par with or better than these existing approaches without requiring any expert labels or demonstrations. Additionally, we report the results of a large-scale qualitative experiment in which STERLING enabled semi-autonomous robot navigation on a 3-mile long hiking trail.

The key contributions of this paper are— 1) *Self-supervised TErrain Representation LearnING* (STERLING), a novel approach that learns terrain representations from easy-to-collect unconstrained robot experiences, 2) Detailed evaluation of STERLING against baseline methods on the task of operator preference-aligned off-road navigation, and 3) A large-scale qualitative experiment of semi-autonomously hiking a 3-mile long trail, demonstrating the effectiveness of STERLING-features.

## 2   Related Work

In this section, we review related work on terrain-aware visual off-road navigation. We specifically focus on approaches that learn to navigate off-road conditions using supervised and self-supervised learning.

### 2.1   Supervised Methods

Several approaches in the past have proposed using supervised learning from large-scale data to navigate off-road environments. We divide them into two categories as follows.

**End-to-End Learning:** The initial success of applying learning-based solutions to off-road terrain-aware navigation was by LeCun et al. [28] who used a convolutional network to learn to drive in off-road conditions. More recently, Bojarski et al. [21] trained a deep neural network end-to-end using several miles of driving data collected on a vehicle in the real world. While both approaches were promising in urban and off-road environments, end-to-end methods require large amounts of data and are well-known to suffer from domain and covariate shifts [29, 30, 31].

**Image Segmentation:** Unlike end-to-end approaches that learn behaviors, segmentation-based approaches seek to characterize terrain using a set of known semantic classes, and the resulting semantic features are consumed by downstream planning and control techniques for navigation [32, 19, 33]. Guan et al. [19] propose GANav, a transformer-based architecture to pixel-wise segment terrains, trained on RELLIS [17] and RUGD [16] datasets, with manually assigned terrain costs. While effective at terrain awareness, segmentation-based methods are fixed to the specific terrain types available in the datasets and require additional labeling effort to generalize to novel terrains. In STERLING, we do not require semantically labeled datasets and learn terrain representations from unconstrained experience collected onboard a mobile robot.

---

[1]A preliminary version of this work was presented at the PT4R workshop at ICRA 2023 [27]

## 2.2 Self-Supervised Learning

To alleviate the need for extensive human labeling, self-supervised learning methods have been proposed to either learn terrain representations or costs from data gathered onboard a mobile robot.

**Representation Learning:** Brooks et al. [34] utilize contact vibrations and visual sensors to classify terrains via self-supervision. Loquercio et al. [35] use proprioceptive supervision to predict extrinsic representations [36] of terrain geometry from vision, used as inputs to drive a Reinforcement Learning-based locomotion policy. In this work, we do not learn a robot-specific locomotion policy and instead learn relevant representations for off-road terrain awareness. Zürn et al. [8] introduce SE-R which utilizes acoustic and visual sensors on the robot to segment terrains using a self-supervised triplet-contrastive learning framework. Using triplet-based contrastive learning methods requires negative samples which may not be available when learning using unlabeled data. In STERLING, we use recently proposed non-contrastive unsupervised learning approaches such as VICReg [37] that do not require any negative samples and instead rely on correlations between data modalities to learn relevant terrain representations.

**Cost Learning:** Several methods have applied self-supervision to assign traversability costs for the downstream off-road navigation task [7, 38, 26, 39, 40, 41, 42]. Specifically, these methods rely on inertial spectral features [7], future predictive models [26], inertial-odometry errors [38], or force-torque values from foothold positions [39, 43] as self-supervision signals to learn a traversability cost map, used to evaluate candidate actions. More recently, Frey et al. [44] have proposed an online traversability estimation approach inspired by the above self-supervision schemes. Instead of inferring costs or rewards using self-supervision for a fixed task, in this work, we focus on learning relevant visual features from unconstrained robot experiences that could be used in downstream tasks. This framework allows a designer to reuse features across tasks without retraining entirely from scratch.

**Hybrid Methods:** The approach closest to ours is VRL-PAP [6] which requires human expert teleoperated demonstrations of a particular trajectory pattern to both explicitly learn visual terrain representations as well as to infer terrain preference costs. However, in this work, we focus on learning terrain features from unconstrained robot experiences without requiring human experts in the field for demonstrations, which is a more general problem than the one considered by VRL-PAP.

## 3 Approach

In this section, we introduce the self-supervised terrain representation learning approach, STERLING, proposed in this work. We first describe the offline pre-processing performed on unconstrained robot data and then summarize the self-supervision objectives. Finally, we describe the problem formulation for preference-aligned off-road navigation and present how features learned using STERLING can be utilized within a planner for terrain-aware and preference-aligned navigation.

### 3.1 Data-Collection and Pre-Processing

STERLING learns terrain representations from unconstrained, unlabeled robot experiences collected using any navigation policy. This policy may be, for instance, non-expert human teleoperation, curiosity-driven exploration [45], or point-to-point navigation using any underlying planner. Compared to requiring a human expert to provide teleoperated demonstrations and labels, collecting this type of robot experience is cheap and easy, thereby providing a scalable pathway to data collection and system improvement. We additionally assume that the robot is equipped with *multiple sensors*, e.g., an egocentric RGB camera, odometry sensors, onboard IMU, proprioceptive, and tactile sensors, that together provide rich multi-modal observations as the robot traverses over different terrains collecting experience. STERLING leverages this multi-modal data by using the correlation between different modalities to inform the learned terrain representations.

In order to learn terrain representations using STERLING, we begin by pre-processing the visual and non-visual observations, which are explained in detail below.

**Visual Patch Extraction:** The egocentric camera observations are homography-projected into a virtual bird's eye view (BEV) frame, assuming that the ground is a flat plane, using the intrinsic and extrinsic camera matrices. As shown in Fig. 1, we project the robot's trajectory onto the BEV frame and extract 64-by-64 pixels (equivalent to the robot's footprint of 0.5-by-0.5 meters) square visual patches of terrain along with the corresponding inertial, proprioceptive, and tactile observations at the same location, along the trajectory. Since the terrain at $s_k$ is unobservable when the robot itself is at $s_k$ (i.e., it is underneath the robot), we extract terrain image patches corresponding to $s_k$ from BEV observations at previous locations $s_{k-1}, s_{k-2}, \ldots$ along its trajectory. Fig. 1 illustrates the offline patch extraction process from two previous viewpoints, however, we extract patches from up to 20 previous viewpoints within 2 meters. Although just one viewpoint is sufficient to learn the correlation between visual and other sensor observations, when planning to navigate, the robot will need to visu-

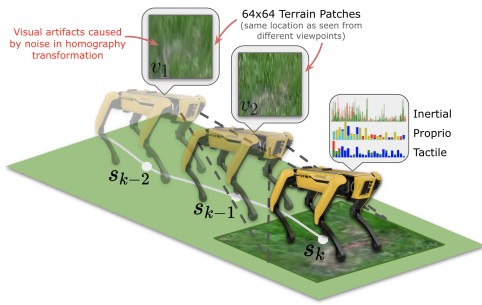

Figure 1: An illustration of the pre-processing performed on unconstrained robot experience. Image patches of traversed terrain at location $s_k$ are extracted from bird's eye view observations at prior locations $s_{k-1}, s_{k-2}$ along the trajectory. The corresponding IPT observations at $s_k$ are transformed from time series to PSD signals. Note the visual artifacts caused by noise in homography transformation from viewpoints farther away from $s_k$.

ally evaluate terrain at future locations, and therefore STERLING also seeks representations that are invariant to patch differences due to viewpoint, also known as *viewpoint invariance*.

**IPT Pre-Processing:** For the inertial, proprioceptive, and tactile (IPT) observations, we retain up to 2-second history and convert the time-series signals into power-spectral density (PSD) representation in the frequency domain. This ensures the IPT time-series data representations used as input to STERLING are invariant to differences in length and phase in the recorded signals. Additional details are provided in Supplementary Section 9.5.

## 3.2 Non-Contrastive Terrain Representation Learning

It is desired for learned representations of terrains to be such that representations of similar terrain are close together in the embedding space and that representations of different terrains are sufficiently far apart. Although we do not possess privileged information such as semantic labels of terrains for training, the visual and kinodynamic observations experienced by the robot reflect similarities and differences between terrain samples. For instance, traversing a smooth terrain that a human may refer to as `cement sidewalk` may lead to relatively smooth motion by the robot's joints, whereas a rough terrain such as what might be referred to as `marble rocks` may correspond to jerkier motion. STERLING leverages this multi-modal experience observed by the robot and computes a correlation objective between visual and inertial-proprio-tactile signals to learn desired terrain representations. Additionally, STERLING uses viewpoint invariance as an objective unique to the visual component of the experience to learn viewpoint-invariant terrain representations.

Fig. 2 provides an overview of the self-supervised representation learning framework adopted in STERLING. A parameterized visual encoder (4-layer CNN with 0.25 million parameters) encodes terrain image patch observations $v_1$ and $v_2$ of the same location $s$ into visual representations $\phi_{v_1}$ and $\phi_{v_2}$, respectively, collectively referred to as $\phi_{v_{1,2}}$ for brevity. Similarly, an inertial-proprio-tactile encoder (4-layer MLP with 0.25 million parameters) encodes frequency domain IPT observations of the robot at that location to an inertial-proprio-tactile representation $\phi_i$. We follow the framework of prior self-supervised representation learning algorithms from the computer vision community such as VICReg [37], and utilize a parameterized projector network (2-layer MLP with 0.25 million

parameters) that maps encoded visual and non-visual representations independently to a higher-dimensional feature space $\psi_{v_{1,2}}$ and $\psi_i$ respectively, over which the self-supervision objectives are computed. The STERLING objective composed of the multi-modal correlation $\mathcal{L}_{MM}(\psi_{v_{1,2}}, \psi_i)$ and viewpoint-invariance $\mathcal{L}_{VI}(\psi_{v_1}, \psi_{v_2})$ objectives are defined as:

$$
\begin{aligned}
\mathcal{L}_{\text{STERLING}} &= \mathcal{L}_{VI}(\psi_{v_1}, \psi_{v_2}) + \mathcal{L}_{MM}(\psi_{v_{1,2}}, \psi_i) \\
\mathcal{L}_{VI}(\psi_{v_1}, \psi_{v_2}) &= \mathcal{L}_{\text{VICReg}}(\psi_{v_1}, \psi_{v_2}) \\
\mathcal{L}_{MM}(\psi_{v_{1,2}}, \psi_i) &= [\mathcal{L}_{\text{VICReg}}(\psi_{v_1}, \psi_i) + \mathcal{L}_{\text{VICReg}}(\psi_{v_2}, \psi_i)]/2
\end{aligned}
\tag{1}
$$

$\mathcal{L}_{\text{VICReg}}$ is the VICReg loss that is composed of variance-invariance-covariance representation learning objectives, as proposed by Bardes et al. [37]. Given two alternate projected representations $Z$ and $Z'$ of a data sample (in STERLING, $Z$ and $Z'$ are projected representations of the visual and non-visual sensor modalities), the VICReg loss is defined as $\mathcal{L}_{\text{VICReg}}(Z, Z') = \lambda s(Z, Z') + \mu[v(Z) + v(Z')] + \nu[c(Z) + c(Z')]$. Note that while Bardes et al. use VICReg to learn representations from visual inputs using artificial image augmentations, in this work, we extend VICReg to multi-modal inputs and use real-world augmentations via multi-viewpoint image patches as described in Sec. 3.1. $\lambda$, $\mu$, and $\nu$ are hyper-parameters and the functions $v$, $s$, and $c$ are the variance, invariance, and covariance terms computed on a mini-batch of projected features. We refer the reader to Bardes et al. [37] for additional details on the individual terms and also define them here for completeness.

The variance term $v$ is a hinge function defined as $v(Z) = \frac{1}{d} \sum_{j=1}^{d} max(0, \gamma - S(z^j, \epsilon))$, where $S$ is the standard deviation, and $d$ is the dimensionality of the projected feature space. $c$ is the covariance term, defined as $c(Z) = \frac{1}{d} \sum_{i \neq j} [C(Z)]_{i,j}^2$, where $C(Z)$ is the covariance matrix of $Z$. $s$ is the invariance term defined as $s(Z, Z') = \frac{1}{n} \sum_i ||z_i - z_i'||$. More details on the individual terms in the loss function are provided in Sec. 9.5. We apply an $l^2$ norm on the visual and non-visual features to ensure they are on a hypersphere, which helped improve the quality of learned representations. On a mini-batch of data containing paired terrain image patches and IPT observations, we compute the $\mathcal{L}_{\text{STERLING}}$ loss and update parameters of the two encoder networks and the shared projector network together using Adam optimizer.

### 3.3 Preference-Aligned Off-Road Navigation

In this subsection, we describe the downstream navigation task of preference-aligned visual navigation that we focus on when evaluating STERLING.

**Preliminaries:** We formulate the task of preference-aligned terrain-aware navigation as a local path-planning problem, where the robot operates within a state space $\mathcal{S}$, action space $\mathcal{A}$, and a deterministic transition function $\mathcal{T} : \mathcal{S} \times \mathcal{A} \longrightarrow \mathcal{S}$ in the environment. The state space consists of $s = [x, y, \theta, \phi_v]$, where $[x, y, \theta]$ denote the robot's position in $SE(2)$ space, and $\phi_v$ denotes the visual features of the terrain at this location. Given a goal location $G$, the preference-aligned navigation task is to reach this goal while adhering to operator preferences over terrains. We assume access to a sampling-based planner, the details of which are provided in Supplementary Sec. 8.

**Learning the preference utility:** Following Zucker et al. [46], we learn the utility function $u : \Phi_v \to \mathbb{R}^+$ using human queries. From the predicted terrain features on data samples in our training set, we cluster the terrain representations using k-means with silhouette-score elbow criterion, and sample candidate terrain patches from each cluster, which is presented to the human operator using a GUI. The human operator then provides a full-order ranking of terrain preferences over clusters, which is utilized to learn the utility function $u(.)$, represented by a 2-layer MLP. While recovering absolute cost values from ranked preference orders is an under-constrained problem, we find that this approximation provided by Zucker et al. [46] works well in practice.

## 4 Experiments

In this section, we describe the experiments performed to evaluate STERLING. Specifically, the experiments presented in this section are tailored to address the following questions:

($Q_1$) How effective are STERLING features in comparison to baseline approaches at enabling terrain awareness in off-road navigation?

($Q_2$) How effective are the proposed STERLING objectives in learning discriminative terrain features in comparison to other representation learning objectives?

We investigate $Q_1$ through physical robot experiments on the task of preference-aligned off-road navigation. We perform quantitative evaluations in six different outdoor environments, and then further perform a large-scale qualitative evaluation by semi-autonomously hiking a 3-mile long off-road trail using preference costs learned using STERLING features. To compare various methods, we use the success rate of preference alignment as a metric. If a trajec-

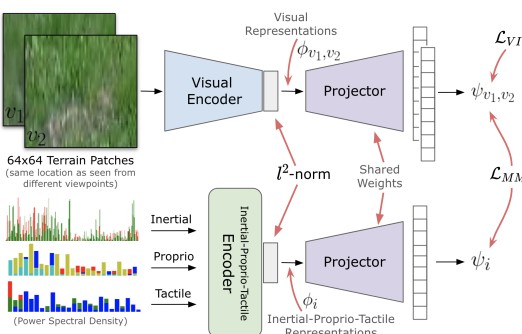

Figure 2: Overview of the training architecture in STERLING. Terrain patches $v_1$ and $v_2$ from different viewpoints of the same location are encoded as $\phi_{v_1}$ and $\phi_{v_2}$ respectively, and mapped into embeddings $\psi_{v_1}$ and $\psi_{v_2}$. Similarly, inertial, proprio, tactile signals are encoded as $\phi_i$, and mapped as $\psi_i$. Self-supervision objectives $\mathcal{L}_{VI}$ for viewpoint-invariance and $\mathcal{L}_{MM}$ for multimodal correlation are computed on the minibatch to perform gradient descent.

tory followed by any algorithm fails to reach the goal, or at any time traverses over any terrain that is less preferred than any traversed by the operator-demonstrated trajectory, we classify the trial as a failure. We additionally investigate $Q_2$ by comparing STERLING against other unsupervised terrain representation learning methods and perform an ablation study on the two STERLING objectives. Additional experiments are provided in Supplementary Sec. 9.2.

**Baselines:** To perform quantitative evaluations for $Q_1$, we compare STERLING with SE-R [8], RCA [7], GANav [19], geometric-only planning [47], and a fully-supervised baseline. SE-R and RCA perform self-supervised learning from unconstrained robot experience to learn terrain representations and traversability costs respectively, making them relevant baselines for this problem. Since there is no open-source implementation of RCA, we replicate it to the best of our abilities. The geometric-only approach ignores terrain costs ($\mathcal{L}_{terrain}$) and plans with geometric cost ($\mathcal{L}_{geom}$) only, making it a relevant ablation on the cost formulation for preference-aware planning. GANav[2] [19] is a segmentation-based approach trained on the RUGD [16] dataset. We additionally train the fully-supervised baseline in which the terrain cost function is learned end-to-end using supervised learning from linear extrapolation of operator preferences. GANav and the fully-supervised baseline require supervision via terrain labels to learn and hence serve as references for comparison. We normalize the terrain cost predicted by all methods to be between 0 and 1 for a fair comparison.

### 4.1 Evaluating Terrain-Awareness via Robot Experiments

In this subsection, we report on experiments to investigate the effectiveness of STERLING features in enabling terrain awareness during off-road navigation. We quantitatively compare the performance of STERLING with baselines RCA [7], GANav [19], SE-R [8] and the fully-supervised baseline, on the task of preference-aligned navigation. We identify six environments within the university campus, with eight different terrain types, as shown in Fig. 3. For this study, we use the same data collected on the robot to train RCA, SE-R, fully-supervised baseline, and STERLING, and the operator

[2]https://github.com/rayguan97/GANav-offroad

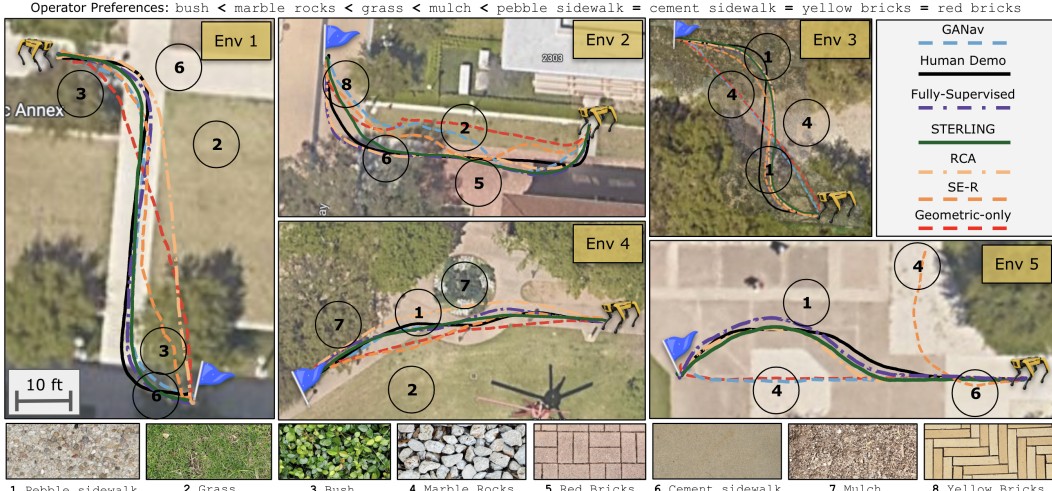

Figure 3: Trajectories traced by different approaches in 5 environments containing 8 different terrains. The operator preferences are shown above. We see that STERLING navigates in an operator-preference aligned manner, by preferring `cement sidewalk`, `red bricks`, `pebble sidewalk`, and `yellow bricks` over `mulch`, `grass`, `marble rocks`, and `bush`, outperforming other baselines and performing on-par with the Fully-Supervised approach.

provides the same rankings for all methods during training. Note that we use the same encoder and utility function across all environments and do not retrain/finetune to each environment to prevent environment-specific overfitting.

Fig. 3 shows the operator's (first author) terrain preferences for all Envs. 1 to 5, and the performance of baseline approaches, including an operator-demonstrated trajectory for reference. In all environments, we see that STERLING navigates in a terrain-aware manner while adhering to the operator's preferences. Note that although Fully-Supervised also completes the task successfully, it requires privileged information such as terrain labels during training, whereas STERLING does not require such supervision, and can potentially be used on large datasets containing unlabeled, unconstrained robot experiences. GANav, trained on the RUGD dataset fails to generalize to unseen real-world conditions. RCA uses inertial spectral features to learn terrain traversability costs and hence does not adhere to operator preference. SE-R does not address viewpoint invariance which is a significant problem in vision-based off-road navigation and hence performs poorly in Envs. 1 and 2. We perform additional experiments in an outdoor environment (Env. 6) to study adherence to operator preferences, detailed in Supplementary Sec. 9.1.

Table 1 shows the success rate of preference alignment for all approaches in all environments, over five different trials. STERLING outperforms other self-supervised baselines and performs on par with the fully-supervised approach. In summary, the physical experiments conducted in six environments quantitatively demonstrate the effectiveness of STERLING features in enabling terrain awareness during off-road navigation.

## 4.2 Evaluating Self-Supervision Objectives

In this subsection, we investigate the effectiveness of STERLING at learning discriminative terrain features and compare with baseline unsupervised terrain representation learning methods such as Regularized Auto-Encoder (RAE) and SE-R [8] and large pretrained networks such as a ResNet-50 pretrained on ImageNet. STERLING uses multi-modal correlation ($\mathcal{L}_{MM}$) and viewpoint invariance ($\mathcal{L}_{VI}$) objectives for self-supervised representation learning, whereas, SE-R and RAE use soft-triplet-contrastive loss and pixel-wise reconstruction loss, respectively. Additionally, we also perform an ablation study on the two objectives in STERLING to understand their contributions to learning discriminative terrain features. To evaluate different visual representations, we perform

unsupervised classification using k-means clustering and compare their relative classification accuracies with manually labeled terrain labels. For this experiment, we train STERLING, SE-R, and RAE on our training set and evaluate on a held-out validation set. Fig. 4 shows the results of this study. We see that STERLING-features using both the self-supervision objectives perform the best among all methods. Additionally, we see that using a non-contrastive representation learning approach such as VICReg [37] within STERLING performs better than contrastive learning methods such as SE-R, and reconstruction-based methods such as RAE. This study shows that the proposed self-supervision objectives in STERLING indeed help learn discriminative terrain features.

| Approach | Environment | | | | | | |
|---|---|---|---|---|---|---|---|
| | **1** | **2** | **3** | **4** | **5** | **6 (a)** | **6 (b)** |
| Geometric-only | 0/5 | 0/5 | 0/5 | 0/5 | 0/5 | 0/5 | 5/5 |
| RCA[7] | 2/5 | 4/5 | 2/5 | 0/5 | 1/5 | 5/5 | 0/5 |
| GANav[19] | 5/5 | 0/5 | 0/5 | 5/5 | 0/5 | 4/5 | 5/5 |
| SE-R[8] | 1/5 | 0/5 | 5/5 | 1/5 | 3/5 | 5/5 | 4/5 |
| Fully-Supervised | 5/5 | 5/5 | 5/5 | 5/5 | 5/5 | 5/5 | 5/5 |
| STERLING (Ours) | 5/5 | 5/5 | 5/5 | 5/5 | 5/5 | 5/5 | 5/5 |

Table 1: Success rates of different algorithms on the task of preference-aligned off-road navigation

## 5 Limitations and Future Work

STERLING requires traversing over terrains in order to learn representations, which may be unsafe in certain situations. Uncertainty-aware safe exploration and exploration focusing on informative and diverse terrains for data collection is a promising direction for future work. Extending STERLING features to work on unstructured non-flat environments such as stairs [48] and boulders [49] is another promising direction for future work. Extending STERLING by pretraining with large-scale off-road datasets using modern architectures such as transformers that are known to scale well with large-scale data is an exciting direction for future work.

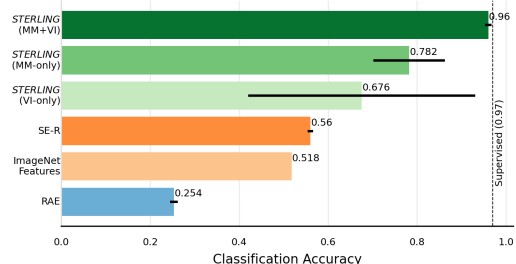

Figure 4: Ablation study depicting classification accuracy (value closer to $1.0$ is better) from terrain representations learned using different approaches and objectives. The combined objective (VI + MM) proposed in STERLING achieves the highest accuracy, indicating that the learned representations are sufficiently discriminative of terrains.

## 6 Conclusion

In this paper, we introduce *Self-supervised TErrain Representation LearnING* (STERLING), a novel framework for learning terrain representations from easy-to-collect, unconstrained (e.g., non-expert), and unlabeled robot experience. STERLING utilizes non-contrastive representation learning through viewpoint invariance and multi-modal correlation self-supervision objectives to learn relevant terrain representations for visual navigation. We show how features learned through STERLING can be utilized to learn operator preferences over terrains and integrated within a planner for preference-aligned navigation. We evaluate STERLING against state-of-the-art alternatives on the task of preference-aligned visual navigation on a Spot robot and find that STERLING outperforms other methods and performs on par with a fully-supervised baseline. We additionally perform a qualitative large-scale experiment by successfully hiking a 3-mile-long trail using STERLING, demonstrating its robustness to off-road conditions in the real world.

**Acknowledgments**

This work has taken place in the Learning Agents Research Group (LARG) and Autonomous Mobile Robotics Laboratory (AMRL) at UT Austin. LARG research is supported in part by NSF (CPS-1739964, IIS-1724157, NRI-1925082), ONR (N00014-18-2243), FLI (RFP2-000), ARO (W911NF19-2-0333), DARPA, Lockheed Martin, GM, and Bosch. AMRL research is supported in part by NSF (CAREER2046955, IIS-1954778, SHF-2006404), ARO (W911NF-19-2- 0333, W911NF-21-20217), DARPA (HR001120C0031), Amazon, JP Morgan, and Northrop Grumman Mission Systems. Peter Stone serves as the Executive Director of Sony AI America and receives financial compensation for this work. The terms of this arrangement have been reviewed and approved by the University of Texas at Austin in accordance with its policy on objectivity in research.

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

# 7 Data Collection

In all experiments, we use a legged Boston Dynamics Spot robot and collect robot experiences on eight different types of terrain around the university campus that we labeled as `mulch`, `pebble sidewalk`, `cement sidewalk`, `grass`, `bushes`, `marbled rock`, `yellow bricks`, and `red bricks`. The data is collected through human teleoperation (by the first and second authors) such that each trajectory contains a unique terrain throughout, with random trajectory shapes. Note that STERLING does not require a human expert to teleoperate the robot to collect robot experience nor does it require the experience to be gathered on a unique terrain per trajectory. We follow this data collection approach since it is easier to label the terrain for evaluation purposes. STERLING can also work with random trajectory lengths, with multiple terrains encountered along the same trajectory, without any semantic labels such as terrain names, and any navigation policy can be used for data collection. We record 8 trajectories per terrain, each five minutes long, and use 4 trajectories for training and the remaining for validation.

# 8 Sampling-based Planning

**Sampling-based planning:** We assume access to a receding horizon sampling-based motion planner with a fixed set of constant-curvature arcs $\{\Gamma_0, \Gamma_1, \ldots, \Gamma_{ns}\}$, $\Gamma \in \mathcal{S}^N$ which solves for the optimal arc $\Gamma^* = \arg\min_{\Gamma}[\mathcal{J}(\Gamma, G)]$, minimizing the objective function $\mathcal{J}(\Gamma, G), \mathcal{J} : (\Gamma, G) \longrightarrow \mathbb{R}^+$. For the task of preference-aligned off-road navigation, we assume the objective function is composed of two components $\mathcal{J}_{geom}(\Gamma, G)$ and $\mathcal{J}_{terrain}(\Gamma)$, and can be defined as $\mathcal{J}(\Gamma, G) = \alpha \mathcal{J}_{geom}(\Gamma, G) + (1 - \alpha)\mathcal{J}_{terrain}(\Gamma)$. $\mathcal{J}_{geom}(\Gamma, G)$ is the geometric cost that deals with progress towards the goal $G$ and avoiding geometric obstacles, whereas $\mathcal{J}_{terrain}(\Gamma)$ is the terrain cost associated with preference-alignment. We utilize the geometric cost as defined in AMRL's graph navigation stack [3]. The multiplier $\alpha \in [0, 1]$ trades off relative contributions of the geometric and terrain preference components of the path planning objective. A 1D time-optimal controller translates the sequence of states in the optimal trajectory $\Gamma^*$ to a sequence of receding horizon actions $(a_0, a_1, \ldots, a_N)$. For a given arc $\Gamma = \{s_0, s_1, \ldots, s_N\}$, such that state $s_0$ is closest to the robot, the terrain-preference cost can be computed as follows.

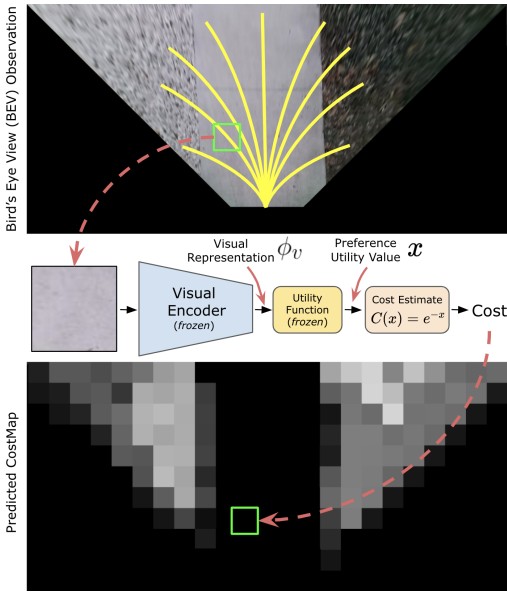

Figure 5: An overview of the cost inference process for local planning at deployment. The constant-curvature arcs (yellow) are overlayed on the BEV image, and the terrain cost $\mathcal{J}_{terrain}(\Gamma)$ is computed on patches extracted along all arcs. White is high cost and black is low cost.

$$\mathcal{J}_{terrain}(\Gamma) = \sum_{v_i \sim \Gamma, i=0}^{N} \frac{\gamma^i C(u(f_v(v_i)))}{N + 1} \qquad (2)$$

The function $f_v(.)$ maps from RGB space of a visual patch of terrain $v_i$ at a specific state $s_i$, to its visual representation $\phi_v \in \Phi_v$. For instance, $f_v$ can be the visual encoder learned using STERLING, as described in Section 3.2. The utility function $u(.)$ maps the visual representation $\phi_v$ of a patch of terrain to a real-valued utility of preferences. We follow the utility function formulation of Zucker et al. [46] and assume the terrain preference cost follows a multiplicative formulation such that given

[3]https://github.com/ut-amrl/graph_navigation

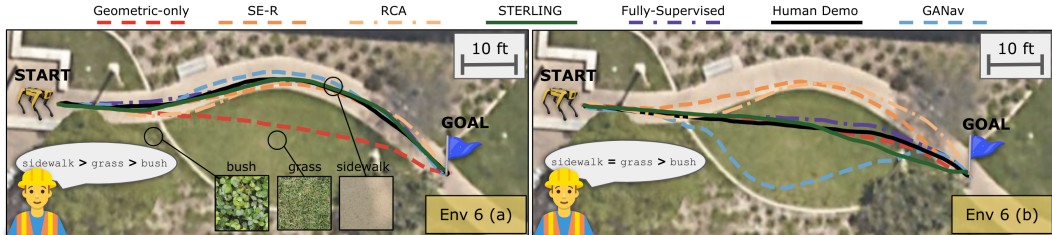

Figure 6: Trajectories traced by different approaches for the task of preference-aligned off-road navigation. Shown here are two different preferences expressed by the operator in the same environment—in 6 (a), sidewalk is more preferred than grass which is more preferred than bush, and in 6 (b), grass and sidewalk are equally preferred and bush is least preferred. We see that without retraining the terrain features, in both cases (a) and (b), STERLING optimally navigates to the goal while adhering to operator preferences.

a utility value $x \in \mathbb{R}^+$, the traversability cost is $C(x) = e^{-x}$. The discount factor $\gamma$ weighs the terrain cost proportional to its proximity to the robot. We set $\gamma$ to $0.8$, which we find to work well in practice.

**Planning at Deployment:** Fig. 5 provides an overview of the cost inference process for local planning at deployment. To evaluate the terrain cost $\mathcal{J}_{terrain}(\Gamma)$ for the constant-curvature arcs, we overlay the arcs on the bird's eye view image, extract terrain patches at states along the arc, and compute the cost according to Eq. 2. We compute the visual representation, utility value, and terrain cost of all images at once as a single batch inference. Since the visual encoder and the utility function are relatively lightweight neural networks with about 0.5 million parameters, we are able to achieve real-time planning rates of 40 Hz using a laptop-grade Nvidia GPU.

## 9 Additional Experiments

In this section, we detail additional experiments performed to evaluate STERLING-features against baseline approaches.

### 9.1 Preference Alignment Evaluation

In addition to the evaluations of STERLING-features with baseline approaches in five environments as shown in Sec. 4, we utilize Env. 6 to further study adherence to operator preferences. We hypothesize that the discriminative features learned using STERLING is sufficient to learn the preference cost for local planning. To test this hypothesis, in Env. 6 containing three terrains as shown in Fig. 6, the operator provides two different preferences 6(a) and 6(b). While bush is the least preferred in both cases, in 6(a), sidewalk is more preferred than grass and in 6(b), both grass and sidewalk are equally preferred. We see in Fig. 6 that using STERLING features, the planner is able to sufficiently distinguish the terrains and reach the goal while adhering to operator preferences. Although SE-R [8] adheres to operator preference in 6(b), it incorrectly maps grass to bush, assigning a higher cost and taking a longer route to reach the goal. On the other hand, RCA [7] fails to adhere to operator preferences since it directly assigns traversability costs using inertial features.

### 9.2 Large-Scale Qualitative Evaluation

In this subsection, we perform a qualitative evaluation of STERLING by reporting a large-scale study of semi-autonomously hiking a 3-mile-long off-road trail using the Spot robot.

We train STERLING using unconstrained robot experience collected within the university campus and train the preference utility function using operator-provided preferences: marble rocks < grass < dirt = cement. The task is to navigate the trail without a global map while adhering to operator preferences at all times. Since we do not use a global map, visual terrain awareness is

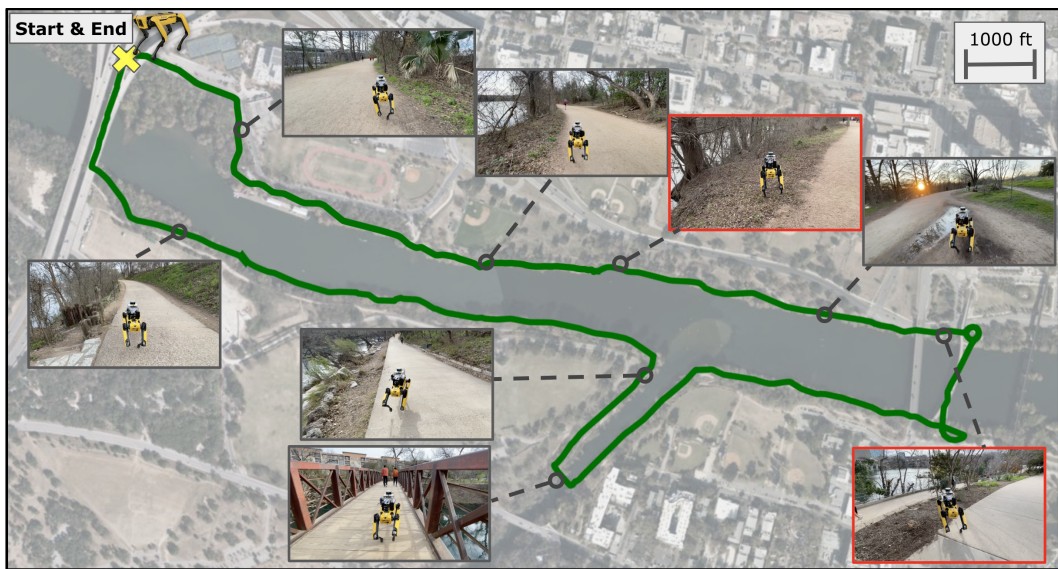

Figure 7: A large-scale qualitative evaluation of STERLING on a 3-mile outdoor trail. STERLING features successfully complete the trail with only two manual interventions (shown in red).

necessary to navigate within the trail and avoid catastrophic events such as falling into the river next to the trail. We set a moving goal of six meters in front of the robot, updated every second. While the robot navigates autonomously, the operator walks behind the robot and takes manual control only to correct the robot's path during forks, or to yield to incoming pedestrians and pets. The attached supplementary video shows the robot navigating the trail successfully while avoiding less preferred terrains. The robot needed two manual interventions while traversing along the trail. Fig. 7 shows the 3-mile trajectory traced by the robot and the two failure cases that required manual intervention. This large-scale qualitative experiment demonstrates the reliability of STERLING during real-world off-road deployments.

### 9.3 Experiments on a Wheeled Mobile Robot

STERLING is intended as a general algorithm to learn relevant terrain representations for off-road navigation. Towards demonstrating the versatility of STERLING to being applied to robots of different morphology, we conduct two additional experiments on the Clearpath Jackal, a wheeled mobile robot.

**Learning Representations on Wheeled Robots:** We utilize unconstrained data collected on the Jackal consisting of multi-modal visual and inertial sensor data and learn terrain representations using STERLING followed by a utility function of operator preferences. Fig. 8 (STERLING-Jackal) shows the path traversed by the Jackal in Env. 6, following the human preference sidewalk > grass > bush. This experiment demonstrates the applicability of STERLING on wheeled robots with inertial sensors, as against legged robots that have access to additional sensors such as joint encoders and tactile information.

**Zero-Shot Cross-Morphology Transfer:** In a noteworthy experiment to evaluate the transferable property of terrain representations across robot morphologies, we utilized the visual encoder trained on data from the legged Spot robot and applied it on the wheeled Jackal robot without additional fine-tuning. Fig. 8 (STERLING-Spot) showcases the Jackal's trajectory, leveraging STERLING representations learned from Spot's data, and adhering to the operator's terrain preference: sidewalk > grass > bush. Fig. 10 shows costmaps generated using STERLING features, used by the sampling-based planner to navigate in an operator-aligned manner. This demonstrates STER-

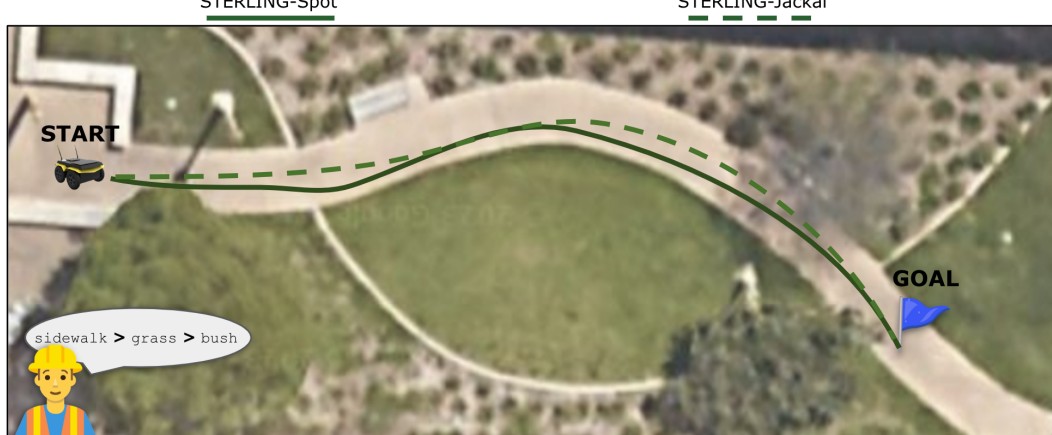

Figure 8: Experimental study of STERLING on a Clearpath Jackal—a wheeled mobile robot in Environment 6. STERLING-Spot shows the trajectory traced using STERLING trained on data collected on the Spot, deployed zero-shot on the Jackal robot, whereas STERLING-Jackal shows the trajectory traced by STERLING trained on data collected on the Jackal, deployed also on the Jackal robot. In both experiments, we see the robot reach the goal successfully while adhering to human operator's preferences over terrains (sidewalk > grass > bush).

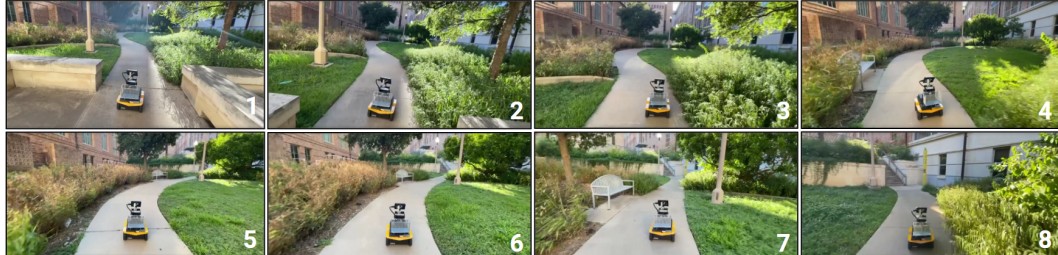

Figure 9: Clearpath Jackal, a wheeled robot navigating using STERLING features trained on unconstrained data collected on the Jackal robot (STERLING-Jackal). We see here in Env. 6 that the robot reaches the goal while adhering to operator preferences Sidewalk > Grass > Bush. This experiment demonstrates the versatility of STERLING in being applied to robots of different morphology.

LING's capability to generalize across diverse robotic platforms, emphasizing its adaptability and broad applicability.

Fig. 9 shows a third-person view of the deployment of STERLING-Jackal in Env. 6. In both experiments above, we see the Jackal robot reaches the goal successfully while adhering to human operator preferences, in a terrain-aware manner, highlighting STERLING's adaptability regardless of robot morphology.

## 9.4 On the Efficacy of Multi-Modal Data Over Vision Alone

While it might seem that visual cues are sufficient for distinguishing terrains, as evidenced in Fig.3, the reality is more complex. Variations in lighting, shadow, color, texture, and other artifacts may lead to inconsistent representations for the same terrain type and can render visually distinctive terrains deceptively similar. For instance, while six different visual patches of the terrain "sidewalk" as shown in Fig. 11 might each exhibit unique visual characteristics because of these variations, they all denote the same terrain category and evoke similar inertial-proprio-tactile (IPT) response on the robot (feel similar to the robot). Solely relying on vision may lead to overlooking underlying commonalities between terrains, resulting in inconsistent terrain representations.

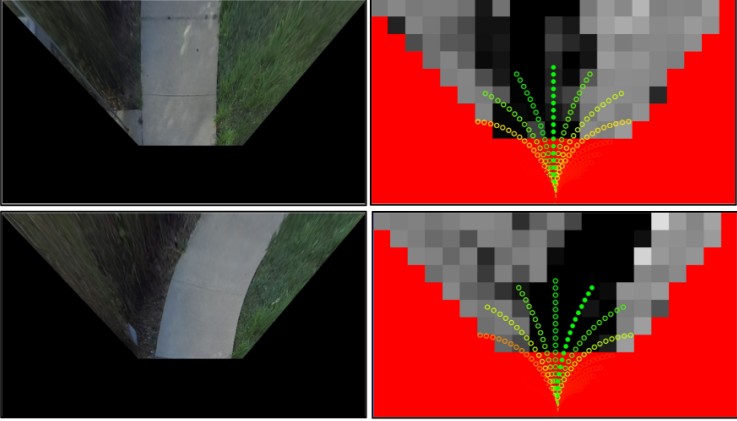

Figure 10: Visualizing the costmaps from the Jackal robot when traversing Env 6., trained using data from the Jackal (STERLING-Jackal).

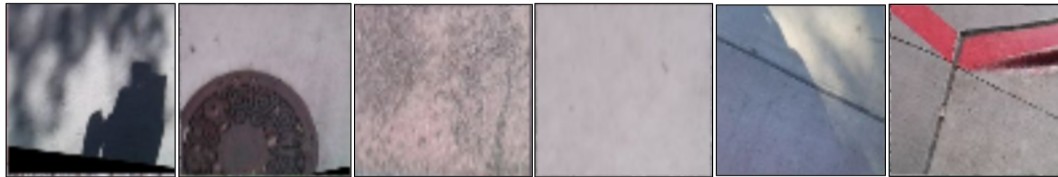

Figure 11: Six distinct instances of `sidewalk` terrain, showcasing the variability in visual appearance due to factors such as lighting, texture, shadows, and other artifacts. Despite the visual differences, each patch represents the same terrain type.

Another concrete example is the scenario of fallen leaves. A sidewalk, a grass patch, and a forest trail could all be covered with fallen leaves, making them visually similar. However, underneath those leaves, the actual terrain properties – and the robot's interaction with them – vary significantly. While the leaves might visually mask the terrain differences, the robot would feel different terrain responses when moving over them due to differences in underlying ground properties.

Furthermore, visual similarity is not a conclusive indicator of identical terrains. Consider four images as a case in point, as shown in Fig. 12. Though the first two and the last two images might seem visually similar, they represent distinct terrains: the first image depicts "bush", the second and third denote "grass", and the fourth is "sidewalk". These three terrains induce different inertial, proprioceptive, and tactile responses in a robot. Thus, the mere semblance of appearance does not capture the relevant features of a terrain.

STERLING's approach of integrating additional modalities allows for more precise terrain identification by accounting for these subtleties. By considering variations and similarities among terrains across different modalities, we ensure relevant terrain representations for off-road navigation. In all examples shown in Figs. 11 and 12, STERLING correctly associates the samples with the right cluster for each terrain.

## 9.5 Experimental Setup and Methodological Details

In this subsection, we outline the specifics of our experimental setup, detailing hyperparameters, architectural decisions, data, and sensory inputs. These insights ensure clarity and reproducibility of our experiments.

In all experiments in STERLING, including the baselines RCA and SE-R, we use a shallow 4-layer CNN with a kernel size of 3 and stride of 1. Our choice of a shallow 4-layer CNN was driven by the specific need for a lightweight and efficient model that could operate in real time at 40Hz

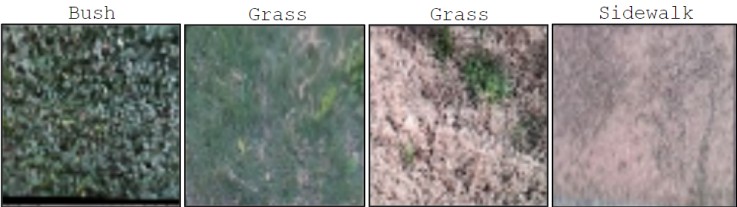

Figure 12: A collection of four terrain patches, illustrating the challenge of terrain representation learning based solely on visual cues. From left to right: bush, grass, grass, sidewalk. Despite visual similarities (and differences), the terrains can elicit different non-visual IPT responses on a robot.

Table 2: Hyperparameter Choices for STERLING Experiments

| Hyperparameter | Value/Range |
|---|---|
| Learning Rate | $3 \times 10^{-4}$ |
| Batch Size | 128 |
| Number of Epochs | 50 |
| Optimizer | Adam |
| Weight Decay | $5 \times 10^{-5}$ |
| Activation Function | ReLU |
| Kernel Size | 3 |
| Stride | 1 |

on a laptop GPU, a requirement that was effectively met with this simple architecture of 0.25 M parameters, which we found was sufficient for the problem. While modern architectures like vision transformers / Mobile-ViT could be applied with larger scale data, the primary concern was real-time performance and compatibility with our robot's hardware. Our experiments and results demonstrate that the selected architecture was sufficient for the purpose, and we do not find evidence that our approach's effectiveness is constrained by this architectural choice.

To train STERLING, SE-R, and RCA, we used a total of 117,604 data samples for all terrains combined. Example raw time-series sensor data is shown in Fig. 14. Each data sample contains a minimum of 2 visual patches and a maximum of 20 visual patches of the same location from multiple different viewpoints from which we randomly sample 2 patches per location during training. We convert the time-series IPT signals into their corresponding Power Spectral Density PSD values. Power spectral density describes the power of a signal across different frequency components. To compute this, we perform a fast-fourier transform over the time-series signal (inertial/proprioceptive/ tactile) and compute the PSD defined as $\text{PSD}(\omega) = \mathbb{E}[|X(\omega)|^2]$ across each frequency component $\omega$.

On the Spot robot, we use a VectorNav IMU to record the inertial signals (angular velocities in the x and y-axis and linear acceleration in z-axis) at 200Hz, the joint angles and velocities of the legged robot, referred as proprioceptive feedback in this work are recorded at 25 Hz, and the feet contact measurements (contact booleans and estimated feet depth from ground) collectively referred to as tactile feedback in this work are recorded at 25Hz. An Azure kinect camera is mounted on the Spot, used for visual sensing of the terrain. On the wheeled Clearpath Jackal robot, we use a Zed2 camera for visual sensing, and utilize the internal IMU sensor for inertial feedback. Fig. 13 depicts the two robots, sensor mounts, and other sensors used in this work.

Note that in all experiments, to prevent overfitting to a specific environment, we pretrain the visual encoder and utility function once and deploy them in all environments. The encoders and utility functions are not being retrained/finetuned per environment, including the large-scale outdoor trail.

**More details on the loss function:** STERLING extends VICReg algorithm initially proposed by Bardes et al. [37] for self-supervised learning from vision-only data. While the foundational work by Bardes et al. uses image augmentations to learn visual representations in a self-supervised way, we utilize images from multiple viewpoints and multi-modal inputs such as vision, inertial, proprio-

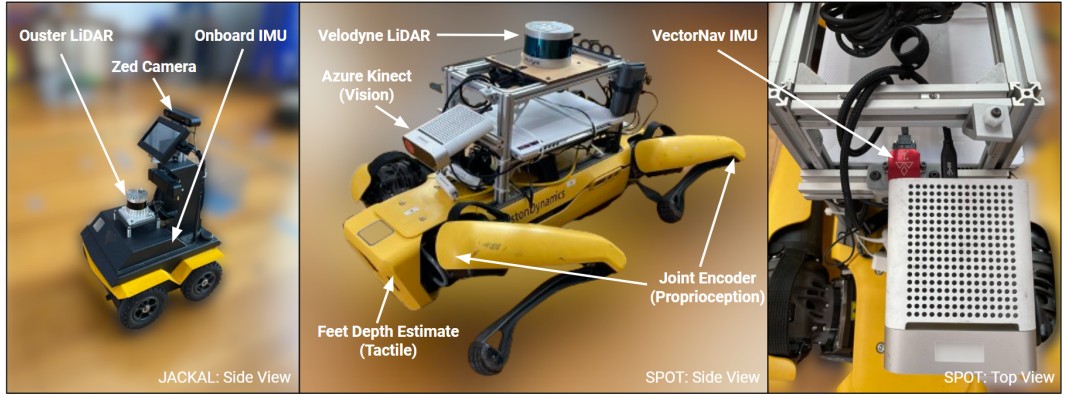

Figure 13: Figure depicting the legged Spot and the wheeled Jackal robot, along with other sensors used in this work.

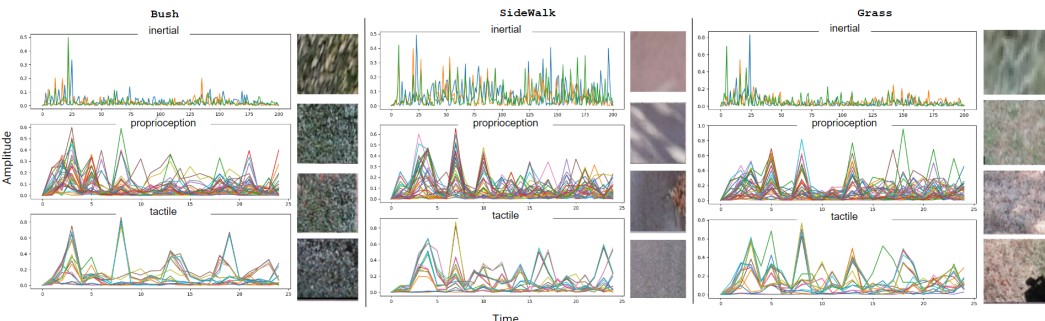

Figure 14: Visual depiction of 1-second of time-series features of inertial-proprioception-tactile data and visual patches of three representative terrains (Bush, SideWalk and Grass).

ceptive, tactile using a novel formulation to learn relevant terrain representations in a self-supervised way. The STERLING loss based on VICReg is defined in Section 3.2. The VICReg loss is defined as $\mathcal{L}_{\text{VICReg}}(Z, Z') = \lambda s(Z, Z') + \mu[v(Z) + v(Z')] + \nu[c(Z) + c(Z')]$, where $\lambda$, $\mu$ and $\nu$ are hyperparameters. We use the values 25.0, 25.0, 1.0 for these hyperparameters respectively, as suggested by Bardes et al. [37]. $s(Z, Z')$ denotes the invariance between the two inputs. In STERLING, this is computed across the two image patches from different viewpoints, and also between the visual and non-visual (IPT) projections. $v(Z)$ denotes the variance across the batch dimension, which we compute for the projections of individual patches and the IPT signals. $c(Z)$ denotes the covariance across the feature dimension which encourages distinct, non-correlative features which we again compute for the projections of individual patches and the IPT signals. We refer the reader to Bardes et al. [37] for additional details regarding individual terms in the loss function. We compute the loss provided in Eq. 1 across a mini-batch of samples and use the Adam optimizer for gradient-based optimization of the visual encoder, IPT encoder and the common projector network.

## 9.6 Visualizing the terrain representations learned using STERLING

Fig. 15 depicts a t-SNE visualization of terrain representations learned using STERLING. Individual patches are color-coded by their ground truth semantic terrain label. We see that STERLING learns relevant features for terrains, given their unique clustering in this latent space.

## 9.7 Visualizing the costmaps

Fig. 16 shows cost visualizations of baseline approaches - RCA [7], SE-R [8], GANav [19] and Fully-Supervised in comparison with STERLING. Fig. 16 shows that RCA and SE-R exhibit issues with

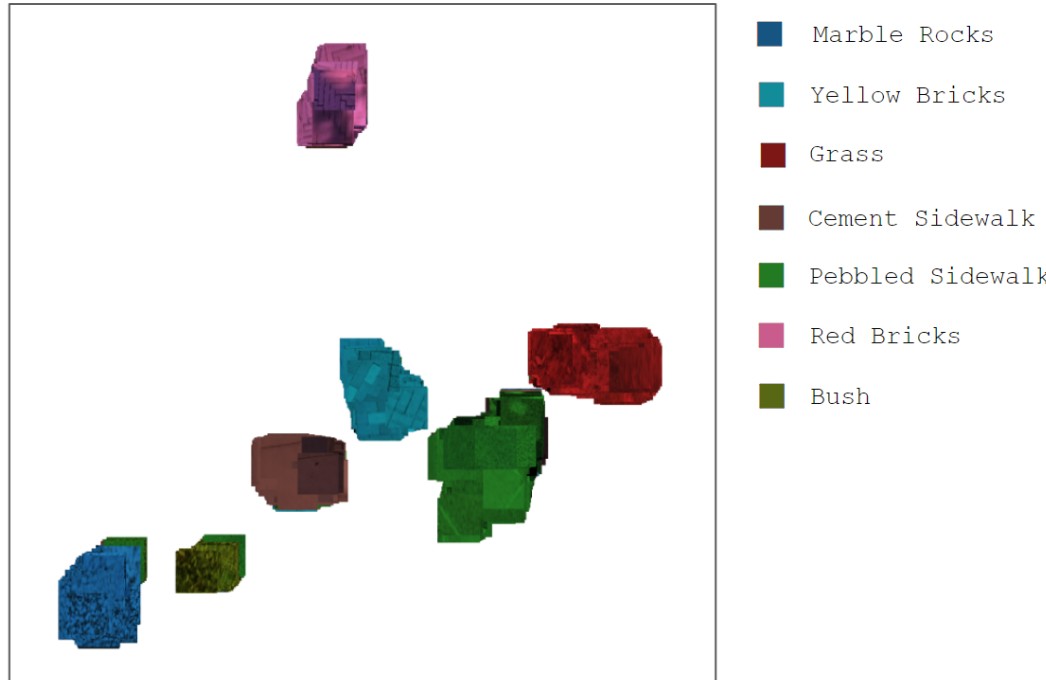

Figure 15: t-SNE visualization of terrain representations learned using STERLING. Each data point represents a terrain example, color-coded by its ground truth label. The clustering of colors show-cases the efficacy of STERLING in capturing meaningful and distinctive terrain features.

visual artifacts due to homography transformations. GANav, trained on the RUGD [16] dataset, fails to generalize to novel real-world situations. In contrast, costmaps from both the fully-supervised model and STERLING efficiently guide planning, as demonstrated by quantitative results in Section 4 and results in behaviors that align with operator preferences, prioritizing sidewalks over terrains like rocks or bushes.

## 9.8 Generalization to Unseen Terrains

During autonomous off-road navigation, generalization to novel terrains is paramount. Although difficult to comment on generalizability, we document an instance during the large-scale deployment where STERLING navigates around an unseen terrain "water puddle", as shown in Fig. 17.

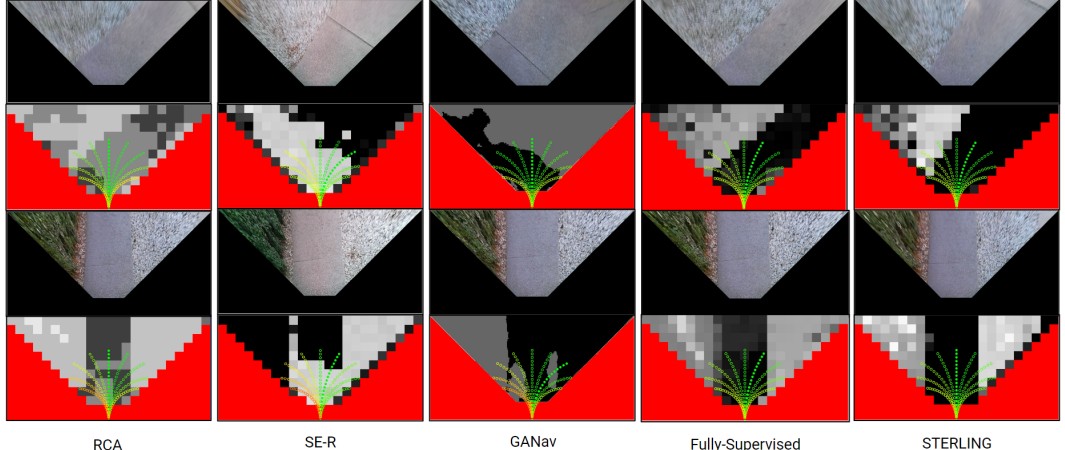

Figure 16: Comparative visualization of the costmaps generated by STERLING (this work) and other baseline algorithms (RCA [7], SE-R [8], GANav [19], Fully-Supervised) for a given scene. Paired with each costmap is a bird's-eye view image of the corresponding terrain. In the costmaps, white regions indicate high traversal cost, black signifies low cost, and areas in red are ignored or non-observable regions. We see that compared to other approaches, using STERLING features results in costmaps that align with operator preference of Sidewalk > Rocks > Bush.

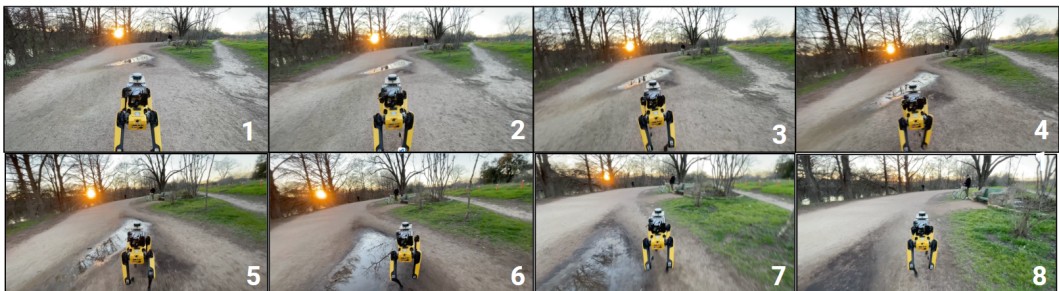

Figure 17: STERLING navigating around an unfamiliar terrain, specifically a "water puddle", during the qualitative 3-mile off-road deployment.

