# OpenReview forum: "STERLING: Self-Supervised Terrain Representation Learning from Unconstrained Robot Experience"
_robot-learning.org/CoRL/2023/Conference — CoRL 2023 Poster_

### Official Review · Reviewer_oD4L · 2023-07-12

**Confidence:** 4
**Originality:** Good
**Technical Quality:** Very Good
**Clarity Of Presentation:** Good
**Impact:** 3

**Recommendation:**

Weak Accept: I recommend accepting the paper, but will not argue for my recommendation if the majority of other reviewers have a different opinion.

**Review:**

## Strength
1. The paper is well-organized and easy to follow.
2. The proposed framework is self-supervised, which reduces the need for manual labeling.
3. With retraining, the network outperforms baselines in sidewalk/brick following planning.
4. The proposed method is deployed and tested on a real robot.

## Weakness
1. The limitations of the proposed work are not well addressed.
2. Some of the terminologies are not well-defined. For example, the term "visual invariance" is repetitively used without referencing what it is invariant to. Is it invariant to $SO(2)$? Or is it invariant to time? Another example is the input IPT data.
3. It seems the user preference is encoded during training. (Because of the preference utility function.) If the user preference changes, a new training process need to be conducted.
4. The main contribution is the self-supervised terrain representation learning module. However, no in-depth evaluation of the terrain representation is provided in the main paper. I recommend replacing the large-scale evaluation with the experiments in Appendix 8.2 and adding more in-depth evaluations on the terrain representation learning part.
5. The network is retrained for each experiment. This reduces the credibility of the results. The network can just be overfitted to a specific tasks/scenarios.
6. The long-distance experiment doesn't give too much insight into the proposed terrain representation network, as it depends on many different factors outside of the contribution of the proposed framework (planning, human intervention, etc.).

Also, see questions for rebuttal.

**Quality Of The Limitations Section:**

Limitations are not well addressed

**Questions For Rebuttal:**

1. The description of the input signals is a bit obscure. Proprioceptive observations often refer to a boarder collection of sensory data, including IMU, joint encoders, kinematics, and contact sensors [1*]. It is difficult to assess what sensory data are being input into the system by "inertial, proprioceptive, and tactile observations." (line 143). I recommend the authors mention the sensors, e.g., joint encoders, force sensors on foot, etc, in the paper.
2. The limitations of the proposed method are not addressed. The one point listed in the limitation section is not the limitation of the proposed work. Please discuss the limitations of the proposed framework.
3. From the paper, it looks like the proposed method is trained separately for different experiments (lines 277, 299, 488). Is it possible that the network is overfitted to specific tasks? For example, since the sidewalk and bricks are the highest priority in experiment 4.1, is it possible that the network only learns to have a low utility cost for sidewalks/bricks and cannot differentiate the rest? What happens if there is no sidewalk/brick in front of the robot? Will the proposed method search for and follow the second preferred terrain?
4. The paper claims the proposed framework allows the Spot to autonomously hike a 3-mile-long off-road trail (line 297). However, it also states, "The operator takes manual control only to correct the robot's path during forks, or to yield to incoming pedestrians and pets." (line 312) In my opinion, it cannot be claimed as "autonomous" if human intervention is required to avoid obstacles and change directions. Furthermore, the paper later reports that only "2" human interventions are needed, but from the video and the map, there're multiple turns and dynamic obstacles (human/pets), which according to line 312, require human intervention.
5. I understand that the lost function is detailed in [26]. However, loss function design is crucial for a self-supervised learning network. I recommend the authors illustrate what the variance, invariance, and covariance function are.
6. In line 138, the paper states, "we extract patches from up to 20 previous viewpoints within 2 meters." However, in Figure 2 and line 174, it seems only 2 images are used as inputs. It would be great if the authors can clarify this.
7. The details of the network hyperparameters are not mentioned. (kernel size, step size, etc.) It'll be difficult for future researchers to reproduce the result if they are not provided.

[1*] Barfoot, Timothy D. State estimation for robotics. Cambridge University Press, 2017.


----------------------------------------------------------------------------------------------------------------------------
Post Rebuttal

I thank the authors for their explanation. The new results in Figure 15 addressed my main concern. As a result, I'm updating my ratings to weak acceptance.

However, the limitation of this work is still not well stated. Instead of simply stating potential future directions, it is more important to point out the limitation of this work. Under what condition does this method fail? What are the angles that are not covered by this method? What are the things this method can/cannot do?

I strongly recommend the authors to address the limitation thoroughly. This can help the future reader to better understand the potential of this paper.

**Robotics Focus:**

Sufficient demonstration on hardware

**Summary Of Paper:**

This paper proposes a self-supervised terrain representation framework, which takes both proprioceptive and image data as inputs. The terrain representation network is combined with a terrain-aware planning scheme proposed by [29] for an off-road navigation task. The experiments using a Boston Dynamics Spot robot show the trained network allows the robot to follow user-preferred terrain during autonomous navigation.

**Summary Of Recommendation:**

The experimental results show promising results on the real robot. However, I'm concerned about the retraining for each experiment, as the network might overfit specific applications instead of learning actual terrain representations. More evidence is needed to support the claims, and there're several issues that need to be addressed. As such, I'm recommending a weak reject.

---

### Official Review · Reviewer_f3rt · 2023-07-16

**Confidence:** 4
**Originality:** Good
**Technical Quality:** Very Good
**Clarity Of Presentation:** Very Good
**Impact:** 3

**Recommendation:**

Strong Accept: I recommend accepting the paper and will argue for my recommendation even if other reviewers hold a different opinion.

**Review:**

**Post-rebuttal comment**
The authors provided a modified version of the paper following reviewer feedback, and I believe it now presents a more convincing claim of STERLING's capabilities for learning generalizable terrain features. I'm confident that most of my original concerns have been addressed in the modified version. I suggest the authors take their time until the final paper submission to ensure good readability as the paper now contains much more details and experimental results than before.  Overall, I am willing to raise my rating from weak accept to strong accept.

Regarding the generalization of STERLING, additional tests on Jackal showed STERLING may generalize to other robot platforms, while the evaluation of the 3-mile test run showed some generalization capabilities to unseen features, while the two interventions may also be seen as failures due to unseen features. The authors should try to mention the two failures to clarify this point in Supplemental Section 9.8.

--
The paper is overall easy to follow and demonstrates impressive results but holds some limitations. See the strengths and weaknesses below.

**Strengths:**
1. The authors were able to use the proposed approach to navigate a 3-mile-long trail with two manual interventions. They quantitatively showed improvement over several baselines in various scenarios as well, which is overall an encouraging demonstration of the effectiveness of the method.
2. Evaluation of STERLING is done with quantitative comparison against GANAV, SE-R, and RCA, which is a diverse set of various prior works on supervised and self-supervised traversability estimation. Against these baselines, STERLING shows better performance in real robot experiments.
3. STERLING relies on self-supervision with some manual preference ranking to generate traversability cost, which is more annotation-efficient than curating annotated datasets. It’s also to be noted that the manual ranking is only done in the context of finetuning STERLING features to a downstream task, and the feature learning itself does not require any annotation. Furthermore, the framework learns more general features that allow downstream tasks compared to prior methods that directly predict traversability.
4. STERLING also does not heavily rely on the actual behavior performed during data collection i.e. it can learn from non-expert demonstrations, and therefore can learn from data that is easier to collect.

**Weaknesses:**
1. Experiments are conducted on relatively simple environments (all six environments are within the university campus with mostly flat terrain). Traversability estimation for off-road navigation would likely require navigating more complex, uneven terrain most of the time. STERLING also uses images projected to BEV with a flat-plane assumption. While this is a reasonable choice in order to focus on the task of preference-aligned navigation, in the context of traversability analysis this is a significantly limiting assumption.
2. One drawback to STERLING is that it focuses on learning from only positive samples i.e. it is only able to learn features from locations it traveled on. It is difficult to comment on the generalizability of the features with respect to new environments and/or new terrain. It seems then that STERLING would require data collection with some prior e.g. novel terrain discovery, and may not perform so well with random experiences.
3. The method requires multi-modal robot experience collected with various sensors, which can be a limitation compared to uni-modal methods such as GANAV.

**Quality Of The Limitations Section:**

Limitations are addressed clearly

**Questions For Rebuttal:**

1. L139: The paper does not describe in detail how image patches from various timestamps are aligned to extract several viewpoints for a particular location $s_{k}$. Is this done by combining depth and odometry?
2. L143: Could the authors provide more detail on power-spectral-density representation or cite relevant work? It seems this preprocessing step may be challenging for other researchers to follow/reproduce if unfamiliar with the topic.
3. Given that the set of input sensor measurements used for STERLING is similar to [19], would it have been possible to include this work as another baseline? (I would suspect this method would fail to capture human preferences similar to RCA).
4. An open question is the generalizability of features learned by STERLING. Would STERLING still be able to perform the evaluated scenarios if its data was less diverse? Based on the information from the supplementary, it seems like the data was collected carefully to ensure the robot traverse over a diverse set of terrains.
5. Sec. 4.1 mainly describes the reasons for the failure of RCA and SE-R. Why does GANAV fail? The method seems to perform fairly well on environments 1, 4, and 6 but fails on the rest. Is the model experiencing a generalization issue?
6. Table 1 shows STERLING surpassing all methods with a 100% succession rate, while the qualitative evaluation required two manual interventions. Could the authors provide details on what these failures were, and whether they are related to the performance of STERLING?
7. Besides uncertainty-aware safe exploration, explorations that focus on discovering diverse and informative trajectories also seem quite important. The data collection process involved teleoperation “such that each trajectory contains a unique terrain throughout,” which perhaps is a strong prior. The authors may consider including this as another potential limitation or discuss this point.
8. Was VICReg chosen as the non-contrastive objective for a specific reason? Could SimSiam[1] or BYOL[2] also be used here? It may be beneficial for authors to briefly describe their basis in choosing VICReg. The authors are also suggested to consider adding definitions of $\lambda, s, \mu, v, c$ from $\mathcal{L}_{VICReg}$ for completeness of the paper.
9. L200 Typo: Sec.6 instead of 7.
10. L219: Authors may consider adding a brief description of what the geometric cost includes.
11. L290: Typo: Sec. 8.1 instead of 9.1.


[1]Chen, Xinlei, and Kaiming He. "Exploring simple siamese representation learning." Proceedings of the IEEE/CVF conference on computer vision and pattern recognition. 2021.

[2]Grill, Jean-Bastien, et al. "Bootstrap your own latent-a new approach to self-supervised learning." Advances in neural information processing systems 33 (2020): 21271-21284.

**Robotics Focus:**

Sufficient demonstration on hardware

**Summary Of Paper:**

In this paper, the authors propose a novel terrain representation learning method based on multi-modal self-supervised learning. Self-supervised TErrain Representation LearnING (STERLING) learns terrain-aware representation with a non-contrastive objective on visual and inertial-proprioceptive-tactile representations extracted from unlabelled robot experience. Features learned by STERLING are evaluated on human preference-aligned navigation, and the experiments along with a qualitative demonstration show that the proposed method outperforms several baselines in navigating through various scenarios with specified preference adherence.

**Summary Of Recommendation:**

The proposed approach STERLING is a novel non-contrastive, non-reconstructive self-supervised learning framework that learns more general representation than prior work for off-road navigation. Comparison of STERLING against several baselines demonstrates the effectiveness of STERLING features for the downstream task of preference-aligned navigation while the qualitative evaluation results are impressive. However, the proposed approach also has some limitations, and the paper has some missing details on analyzing experiment results as well as the method’s overall limitations. Concerns raised by the reviewer are described in Questions for Rebuttal. Lastly, the authors are suggested to consider a code release, and/or adding details on the methodology for reproducibility.

---

### Official Review · Reviewer_jqM5 · 2023-07-19

**Confidence:** 4
**Originality:** Good
**Technical Quality:** Good
**Clarity Of Presentation:** Very Good
**Impact:** 4

**Recommendation:**

Weak Accept: I recommend accepting the paper, but will not argue for my recommendation if the majority of other reviewers have a different opinion.

**Review:**

Strengths

- Using VICReg for learning terrain representation is clever and makes a lot of sense, given that the strength of VICReg is that it can learn from multi-modal inputs.
- The experiment is thorough with comparison to multiple baselines.
- The paper is well-written and easy to follow.

Weaknesses

- The paper misses some important details.
    - Since the approach is about using multiple sensor modalities, it is important to visualize the different modalities and discuss how they are correlated for different types of terrains. Visualizing the embedding space would also be helpful.
    - There is no visualization of the costmap. There is one figure that shows the BEV RGB projection and the costmap in the supplementary, but it will be more informative to compare the costmaps produced by different methods. This would help the readers to better appreciate the effectiveness of the proposed approach.
- The proposed approach uses a rather simple CNN for processing images. It is unclear how a modern vision backbone would change the results.

Comments:
- The proposed idea seems like a good approach for self-supervised terrain representation learning from multiple sensors. My main concern is that it uses a very outdated vision backbone and the terrains being visually distinctive, thus making the effectiveness of multi-modal learning less apparent. More visualization, discussion and analysis could help with strengthening the paper.
- Some of the ablation results in supplementary are important for understanding the contribution of each component. I suggest moving them into the main paper.
- Line 291 references Sec 9.1 in the supplementary, but I cannot find it.

**Quality Of The Limitations Section:**

Limitations are addressed clearly

**Questions For Rebuttal:**

- Why does vision alone not produce good enough terrain features? In Figure 3, it seems that different terrains have quite distinctive appearances.
- The model uses a shallow CNN. Is there any reason not to use a better vision backbone? In the supplementary, it mentions that a laptop GPU is used for inference, which should be able to run modern neural nets in real time. This may also be related to the first question about why vision alone does not produce good terrain features.
- Since the proposed approach requires collecting paired sensor data, it cannot leverage large amounts of image data that is readily available. I wonder how a vision-only model would work in this context. For example, using features from DINO, how much accuracy can you achieve? And how much multi-modal sensor data would help? This does not diminish the benefit of using multi-modal sensor data, but it would help us to understand where multi-modal sensor is helping.
- SE-R does not address viewpoint invariance. Would it be possible to add viewpoint invariance to SE-R? Would it perform comparably to STERLING?
- The supplementary mentions that 4 trajectories for training, each being 5 minutes long. How many total frames were used for training? It will be good to show the four trajectories along with the example sensor data as well.
- When you compare with the baselines, did you use the same network architecture and training data?

**Robotics Focus:**

Sufficient demonstration on hardware

**Summary Of Paper:**

This work proposes a method that enables a robot to learn terrain representation in a self-supervised fashion from unconstrained experience. The key idea is to leverage multi-modal sensor data (color, proprioceptive and tactile) for representation learning. It uses VICReg framework that learns a joint embedding over multiple modalities. For planning, it learns a utility function that maps embeddings to costs from human preferences, and then combines it with a geometric cost with a receding horizon planner. Experiments show that the proposed approach adheres to human preferences better than the baselines, while not requiring expert demonstrations during training.

**Summary Of Recommendation:**

This seems a promising approach of learning terrain representations using non-expert data. The experiments are thorough and convincing. I vote for weak accept. But it would be great if the authors could address my questions since they are important to understand the effectiveness of the approach on real data.

---

### Official Review · Reviewer_DHLH · 2023-07-27

**Confidence:** 5
**Originality:** Poor
**Technical Quality:** Poor
**Clarity Of Presentation:** Very Good
**Impact:** 3

**Recommendation:**

Weak Reject: I recommend rejecting the paper, but will not argue for my recommendation if the majority of other reviewers have a different opinion.

**Review:**

Strengths:
- The paper is mostly clear and easy to read.
- The tackled problem is of relevance and active in the legged robots community

Weaknesses:
- The paper has failed to position itself in the legged robotics state of the art. [A] [B] and more recently (no mandatory as it was arXiv by the CORL submission date [C]).
- The chosen baselines where not targeted to legged robots ([A] and [B] must be included, and why not using [28] as well? ).
- The 3 listed contributions (1. Sterling, 2. evaluations, 3. large scale) get diminished without the proper comparisons.

Legged systems:
[28] Wellhausen L, Dosovitskiy A, Ranftl R, Walas K, Cadena C, Hutter M. Where should i walk? predicting terrain properties from images via self-supervised learning. IEEE Robotics and Automation Letters. 2019 Jan 27;4(2):1509-16.
[A] Wellhausen L, Ranftl R, Hutter M. Safe robot navigation via multi-modal anomaly detection. IEEE Robotics and Automation Letters. 2020 Jan 20;5(2):1326-33.
[B] Loquercio A, Kumar A, Malik J. Learning visual locomotion with cross-modal supervision. In2023 IEEE International Conference on Robotics and Automation (ICRA) 2023 May 29 (pp. 7295-7302). IEEE.
[C] Frey J, Mattamala M, Chebrolu N, Cadena C, Fallon M, Hutter M. Fast Traversability Estimation for Wild Visual Navigation. arXiv preprint arXiv:2305.08510. 2023 May 15.

**Quality Of The Limitations Section:**

Limitations are addressed clearly

**Questions For Rebuttal:**

No questions, see weaknesses above.

**Robotics Focus:**

Sufficient demonstration on hardware

**Summary Of Paper:**

The paper proposes "STERLING" to allow terrain understanding for autonomous off-road navigation without relying on expensive labeled data, engineered features, or expert human demonstrations. The approach takes unlabelled robot experience to learn relevant terrain representations. Physical robot experiments have been carried out to show performance similar to fully-supervised methods and outperform other chosen baselines.

**Summary Of Recommendation:**

Without positioning and comparing with the relevant work in legged robotics the main contribution can not be assessed.

**Post rebuttal **

I thank the authors for the big effort during the rebuttal. And I raise my score to reflect that.

However, I am still on the rejection side as I am not convinced with major points:
1. More general than legged robots: then why to use the  “feet ground depth penetration estimate" for spot?
2. Continual Learning: several times the authors have used this term as if this is a trivial thing to do. There is a vast amount of work in this topic that shows how is easier said than done.

---

### Author Response · Authors · 2023-08-14
**General Summary**

We express our gratitude to the reviewers for their careful consideration of our work. The positive feedback and recognition of novelty and relevance is appreciated. Their suggestions have helped strengthen the revised paper, with additional robot experiments and clarified sections. We particularly thank reviewers **jqM5** and **f3rt** for acknowledging that the revised paper has convinced them that STERLING leads to learning a more robust representation than the baselines.

 - **Reviewer DHLH** appreciated the relevance of this problem to the community and the readability of our work.
 - **Reviewer jqM5** emphasized the thoroughness of our experiments with multiple baselines, and the “clever” use of VICReg by extending it to multi-modal representation learning for off-road navigation. The potential of our work to have a major impact on robotics or machine learning was also highlighted.
 - **Reviewer f3rt** commended our “impressive results” with a diverse selection of baselines, additionally highlighting STERLING’s capacity to generalize across diverse terrains and robot platforms.
 - **Reviewer oD4L** acknowledged STERLING’s self-supervised capability reducing labeling costs, with particular emphasis on its real-robot hardware demonstration.

We believe these positive sentiments underscore the potential and significance of our work. Based on the feedback, we have performed additional experiments and made changes to the paper regarding formatting, and included more details, as summarized below:

**Experiments:**

1) Experiments on a different robot: We perform two additional experiments on the ClearPath Jackal (a wheeled mobile robot). First, we train and deploy STERLING on data collected on the Jackal, showcasing the general applicability of STERLING irrespective of robot morphology. Secondly, we perform a cross–morphology transfer of STERLING trained using data collected on the legged Spot robot and transferred zero-shot on the wheeled Jackal robot.
These additional real-world experiments and their results are summarized in the updated paper Supplemental Section 9.3
2) Generalization: On the large-scale 3-mile qualitative experiment, we included examples where STERLING features generalize to unseen terrains such as “rainwater puddle”, included in Supplemental Section 9.8.

**Text Changes:**


1) Additional experiments on the ClearPath Jackal robot are summarized in Supplemental Section 9.3.
2) We have contrasted STERLING to related work as pointed out by reviewer DHLH, in related work Section 2.2.
3) We have now included a table showing hyperparameter configurations, with a detailed methodology section (Supplemental Section 9.5) for ease of reproducibility, as pointed out by reviewer oD4L.
4) We have included additional details on the loss functions for completeness in Supplemental Section 9.5, as suggested by reviewer oD4L.
5) We have included the ablation experiments on the learned representations in the main paper under Section 4.2, and moved the large-scale qualitative experiments to supplemental Section 9.2, as suggested by reviewer oD4L. We have additionally included visualizations of the learned representations in STERLING in Supplemental Section 9.7, as requested by reviewer f3rt.
6) Following the suggestion of reviewer jqM5, we have included images of costmap produced by STERLING and other baseline methods for comparison in Supplemental Section 9.7. We additionally included costmaps produced by STERLING on the Jackal robot in Supplemental Section 9.3.
7) Finally, we have included a discussion in Supplemental Section 9.5, regarding our proposal of using multi-modal data to learn relevant terrain representations, based on the insightful discussion with reviewer jqM5.

We're confident that these changes amplify the paper's value and thank the reviewers for steering these improvements. We look forward to your final consideration.

---

### Decision · Program_Chairs · 2023-08-30

**Decision:**

Accept (Poster)

**Comment:**

This paper addresses a relevant robotics problem and, after engaging positively with the reviewers, is persuasive in its arguments.  The authors should consider textual revisions to the paper to improve readability, as their (impressive) responsiveness to reviewers has left the work technically dense and not easy to digest.